# Reliability of Near-Infrared Spectroscopy with and without Compression Tights during Exercise and Recovery Activities

**DOI:** 10.3390/sports11020023

**Published:** 2023-01-18

**Authors:** Brett Biddulph, John G Morris, Martin Lewis, Kirsty Hunter, Caroline Sunderland

**Affiliations:** 1Sport, Health and Performance Enhancement (SHAPE) Research Centre, Department of Sport Science, Nottingham Trent University, Nottingham NG11 8NS, UK; 2Qualisys AB, 411 05 Gothenburg, Sweden

**Keywords:** tissue saturation index, NIRS, muscle oxygenation, muscle deoxygenation, compression, reliability

## Abstract

Near-infrared spectroscopy (NIRS) is widely used in sports science research, despite the limited reliability of available data. The aim of the present study was to assess the reliability of NIRS with and without compression tights. Thirteen healthy active males, (age 21.5 ± 2.7 years, body mass 82.1 ± 11.2 kg, BMI 24.6 ± 3.2 kg·m^−2^) completed four trials (two control trials and two trials using compression tights) over a 28-day period. During each trial, participants completed 20 min each of laying supine, sitting, walking (4 km·h^−1^), jogging, and sitting following the jogging. An NIRS device was attached to the muscle belly of the vastus lateralis and gastrocnemius and recorded tissue saturation index (TSI), muscle oxygenation, and muscle deoxygenation. Systematic bias and 95% limits of agreement (LOA) and coefficient of variation (CV) were used to report reliability measures for each activity type. For TSI, systematic bias (LOA) at the gastrocnemius during the control and tights trial ranged from −0.4 to 1.7% (4.4 to 10.3%) and −1.9 to 3.5% (8.1 to 12.0%), respectively. For the vastus lateralis, the systematic bias (LOA) for the control trial ranged from −2.4 to 1.0% (5.1 to 6.9%) and for the tights trial was −0.8 to 0.6% (7.0 to 9.5%). For TSI, the CV during the control trial ranged from 1.7 to 4.0% for the gastrocnemius and 1.9 to 2.6% for the vastus lateralis. During the tights trials, the CV ranged from 3.0 to 4.5% for the gastrocnemius and 2.6 to 3.5% for the vastus lateralis. The CV for muscle oxygenation during the control and tights trials for the gastrocnemius was 2.7 to 6.2% and 1.0 to 8.8% and for the vastus lateralis was 0.6 to 4.0% and 4.0 to 4.5%, respectively. The relative reliability was poorer in the tights trials, but if the aim was to detect a 5% difference in TSI, NIRS would be sufficiently reliable. However, the reliability of muscle oxygenation and deoxygenation varies considerably with activity type, and this should be considered when determining whether to employ NIRS in research studies.

## 1. Introduction

Increased participation in sport has led to individuals examining, and investing in, interventions to enhance performance and recovery. Compression garments are one such intervention that have become widely utilised at all levels of sport [1]. However, research is warranted to assess their potential efficacy for use during performance and recovery [2]. Compression garments include tights, socks, stockings, calf guards, vests, and sleeves. In clinical use, compression tights have been shown to aid blood flow and oxygenation at a local muscular level [3,4]. However, research relating to the benefits of compression in healthy sports people remains equivocal [1,5], and thus rigorous methods to assess the potential benefits of compression garments is needed, particularly if mechanisms of performance and recovery benefit are to be determined.

The application of compression garments reduces venous pooling, blood flow, and lactate during recovery and muscle soreness following muscle damaging exercise [6,7,8,9], and enhances lactate removal during recovery and strength and power recovery following exercise-induced muscle damage [8,10]. However, contradictory research has reported no physiological differences compared with a control when wearing compression garments during exercise and recovery [11,12,13,14]. Thus, further research is warranted to examine the physiological responses elicited, particularly blood flow and muscle oxygenation, when wearing compression garments for potential performance and recovery benefits.

Near-infrared spectroscopy (NIRS) measures relative changes in oxy- and deoxyhaemoglobin and can be used during both passive and exercise activities, allowing muscle oxygen consumption and blood flow to be determined [15]. NIRS can therefore be applied to studies examining physiological responses to compression clothing to determine blood flow and oxygenation. However, to be used during exercise compression studies, the reliability of NIRS must be determined as reliability appears to be specific to the exercise type and intensity.

Studies examining the reliability of NIRS for determining muscle oxygenation have produced equivocal findings, with poorer muscle oxygenation reliability during higher intensity activities. Balas et al. [16] assessed reliability by examining muscle oxygenation in the forearm during rest and handgrip contractions. Tissue saturation index (TSI) and total haemoglobin (tHb) coefficient of variation values were higher for intense activity, ranging from 17.2 to 41.8% in comparison to 8.3 to 12.9% during rest [16]. Similarly, studies that have examined the reliability of NIRS measures across a range of voluntary contractions (10% to 100% MVC) have shown reliability to vary, with ICC values ranging from 0.02 to 0.96 and CV from 1.5 to 36% [17,18,19,20]. Similarly, during co-coordinative exercise activities, such as cycling and running, reliability has been shown to be variable, with CV for muscle oxygenation, tHb, and TSI ranging from 6.1 to 43.5% [21,22,23]. This may be related to the rapid changes in haemodynamics and blood flow that occur during exercise using co-ordinated limb movements [23]. In addition, applying pressure to the muscle and NIRS optode, which would occur when wearing compression garments, can alter the overall reliability of NIRS and must be evaluated [24]. Variations in pressure applied to the NIRS device can alter the pathlength between the muscle and the optode, subsequently impacting the signal quality [24]. Therefore, evidence suggests reliability is highly variable and to date no studies have assessed reliability when wearing a compression garment which may have an additional influence on reliability.

Therefore, the aim of the present study was to assess the between-day reliability of NIRS for the determination of muscle oxygenation, deoxygenation, and tissue saturation index at the gastrocnemius and vastus lateralis with and without compression tights. Reliability was determined during a range of activities that may be completed during recovery from, and travel after, exercise, specifically, supine and seated rest, walking, and slow jogging. Reliability was also determined for a range of cardiovascular measurements, specifically, heart rate, blood pressure, and pulse oxygen saturation.

## 2. Materials and Methods

### 2.1. Participants

Thirteen healthy, active males (age 21.5 ± 2.7 years, body mass 82.1 ± 11.2 kg, BMI 24.6 ± 3.2 kg·m^−2^) completed four experimental trials across a 28- day period. Institutional ethical approval was obtained from Nottingham Trent University (approval number 425). Participants were briefed and completed written consent and health declaration forms. Participants were asked to refrain from strenuous exercise, caffeine, and alcohol 24 h before each trial. Individuals were required to monitor diet for 24 h via recorded diary prior to their first main trial and replicate this diet prior to the remaining trials.

### 2.2. Experimental Trials

In order to minimise the training and detraining effect, four experimental trials were completed within a 28-day period. Two control and two compression trials were randomly assigned for each participant via Latin square. During the control trial, sporting attire was worn (loose shorts and t-shirt) whereas during the compression trial, custom-made compression leggings (Kurio Compression Ltd., Clipstone, UK) and a loose-fitting t- shirt were worn. The experimental protocol was designed to replicate activities that might be used during recovery from strenuous exercise. Each protocol comprised of 20 min of laying supine, sitting, walking, jogging, and sitting (Figure 1). For the jogging aspect, each participant selected a speed at which they would complete a recovery jog (7.0 ± 0.9 km·h^−1^), and this was replicated for the remaining trials. Immediately following the 20 min jog, participants sat for 20 min to determine any changes following the jogging (Figure 1). Prior to any main trials, a familiarisation session was completed where participants were measured for custom-made garments and participants were accustomed with all the equipment used and measurements being made (NIRS, pressure measures, skin fold measurements). Under each condition, the participant was asked to remain quiet throughout and minimise movement when appropriate.

### 2.3. Compression Tights

Compression tights were manufactured by Kurio (Kurio Recovery, Kurio 3D Compression Ltd., Edwinstowe, Nottingham, England). Each participant was individually measured, including limb lengths, circumferences, and waist and hip measurements, and compression tights were custom-made for each participant’s lower-limb geometry, creating similar lower-limb compression profiles between left and right limbs and between participants.

### 2.4. Compression Measurements

Pressures exerted by the compression tights were recorded at the beginning of each activity (Kikuhime Pressure Monitor Medi-Group, Melbourne, Australia). Pressure readings were taken at six specific locations on the opposite limb to which the near-infrared spectroscopy was placed. The six locations were at the ankle bone (a), 5 cm proximal to landmark a (b), on the medial aspect of the maximal calf girth (c), on the anterior aspect of the thigh 10 cm below landmark e (d), the mid-point between the inguinal crease and the superior–posterior border of the patella (e), and 5 cm proximal to landmark e (f) [25].

### 2.5. Physiological and Perceptual Measurements

Heart rate (Polar S810i heart rate monitor, Polar Electro Oy, Kempele, Finland) and pulse oximetry (ANP 100 Finger Pulse Oximeter, Anapulse, Surrey, UK) were recorded every 5 min during each activity. Blood pressure (Omron M2 Basic Blood Pressure monitor, Omron Corporation, Kyoto, Japan) was recorded at the end of each 20 min activity. Rating of perceived exertion (RPE) was recorded every 5 min during the walking and jogging activities [26].

### 2.6. Near-Infrared Spectroscopy Measurements

Muscle oxygenation was measured using 2 portable NIRS devices with optode distances of 30, 35, and 40 mm (Portalite, Artinis Medical Systems BV, Elst, The Netherlands). Following cleaning and shaving, one NIRS device was attached to the muscle belly of the vastus lateralis, 15 cm proximal to the patella head and 4 cm lateral, and a second NIRS device to the gastrocnemius at maximal calf girth. NIRS was attached with double-sided discs and tape and covered with a black bandage (Coban, 3M, UK) to eliminate background light. The position was marked with a permanent pen for subsequent visits. Skinfold thickness was measured using callipers (Harpenden Ltd., UK) at the sites of the NIRS attachment during the familiarisation visit. A modified version of the Beer–Lambert Law, using 2 continuous wavelengths of 760 nm and 850 nm and a differential pathlength factor (DPF) value of 4.95 [27], was used to determine changes in oxyhaemoglobin (HbO_2,_ µM), deoxyhaemoglobin (HHb, µM), and total haemoglobin (tHb) [tHb = HbO2 + HHb]. Tissue saturation index (TSI) was also determined, which represents the absolute measure of oxygenated haemoglobin (TSI = [O2Hb]/([O2Hb] + [HHb])∗100%).

NIRS data were acquired at 10 Hz and were continuously monitored through a Bluetooth connection and instantly uploaded to the device’s software (Oxysoft software, Artinis Medical Systems BV, Elst, The Netherlands). At the start of the trials, the participant was asked to sit for 5 min, with the baseline taken as the final 60 s of the 5 min seated rest period.

### 2.7. Data Processing for NIRS

MATLAB (MATLAB and Statistics Toolbox R2017a, The MathWorks, Inc., Natick, Massachusetts, United States) was used to apply a 4th-order Low-Pass Butterworth filter to the raw NIRS data with a low-pass cut-off frequency of 0.2 Hz to remove high-frequency noise [28]. The cut-off frequency was determined by evaluating a power spectral analysis of data in the frequency domain. TSI was measured in addition to changes from baseline, which were calculated for oxyhaemoglobin (HbO2), deoxyhaemoglobin (HHb), and total haemoglobin (tHb). For each 20 min period, data were averaged per minute, thus producing 20 data points per activity.

### 2.8. Statistical Analyses

Data are presented as mean ± SD. The reliability of the NIRS for the assessment of muscle oxygenation was examined using systematic bias and 95% limits of agreement [29] and coefficient of variation (CV %) based on differences and intra-class correlation (ICC) for the control and compression condition. CV and ICC were calculated using an available spreadsheet [30]. A paired t-test was used initially for physiological measures to assess any significant differences (*p* < 0.05). Reliability measures for the ICC were categorised as: >0.75, good; 0.40 to 0.74, moderate; and <0.40, poor [31]. Statistical analysis was performed using SPSS version 24.0 (SPSS Inc., Chicago, IL, USA).

## 3. Results

Temperature and humidity were not different between trials (mean of all trials: 19.2 ± 2.8 °C, *p* = 0.8; RH: 41.9 ± 11.9%, *p* = 0.05).

### 3.1. Compression Pressure

Compression pressure across all landmarks was similar between the two compression garment trials (*p* > 0.05). Coefficient of variation ranged from 1.0 to 3.6% for all landmarks. Compression readings for individual fitted garments ranged from 7 to 19 mmHg. The highest compression readings were at the gastrocnemius (19 mmHg, location c) and the lowest at the thigh (location f).

### 3.2. Anthropometric Measurements

There was no difference in body mass for the control (81.2 ± 11.06 kg, *p* = 0.72) and compression tights trials (82.3 ± 11.49 kg, *p* = 0.13). Skin fold thickness and girth were similar between trials (all *p* > 0.05, Table 1).

### 3.3. Heart Rate, RPE, and Blood Pressure

Heart rate was similar between the control and compression trials, respectively (*p* > 0.05). Systematic bias was −0.5 to 0.2 beats·min^−1^ (LOA: −19.3 to 33.3 beats·min^−1^) during the control trial and −0.5 to 0.0 beats·min^−1^ (LOA: −11.1 to 19.2 beats·min^−1^) in the compression trials. ICC values for heart rate ranged from 0.56 (moderate) to 0.76 (good) for the control condition and 0.57 (moderate) to 0.77 (good) for the compression reliability. Coefficient of variation (%) was <7.6% for both control (C) and tights (CT) (Supine: C = 6.4%, CT = 5.5%; Sitting: C = 4.0%, CT = 6.5%; Walking: C = 5.8%, CT = 7.2%; Jogging: C = 6.9%, CT = 7.5%; Sitting: C = 7.2%, CT = 5.3%). Rating of perceived exertion during walking was 6.1 ± 0.3 and 6.3 ± 0.8 (*p* = 0.58, CV = 0.2%) in the control condition and 6.1 ± 0.3 and 6.0 ± 0.0 (*p* = 0.34, CV = 0.1%) for the compression tights. Jogging RPE was 7.8 ± 1.4 and 8.0 ± 1.5 (*p* = 0.76, CV = 0.8%) during control and 8.9 ± 2.5 and 8.1 ± 1.5 (*p* = 0.25, CV = 1.3%) whilst wearing compression tights. Blood pressure was similar during each activity in the control and compression trials (*p* > 0.05).

### 3.4. Pulse Oximetry

Oxygenation at the finger was similar throughout the control and compression trials (*p* > 0.05) and demonstrated very good reliability. Systematic bias was −0.2 to 0.2% (LOA: −5.4 to 5.5%) during the control trials and 0.01 to 0.5% (LOA: −4.2 to 4.2%) for compression tights. Coefficient of variation (%) was 0.8 to 2.8% during the control condition and 0.1% to 2.0% for the compression tights trials.

### 3.5. Tissue Saturation Index

TSI reliability was moderate-to-good for the control trial (ICC: 0.56 to 0.96) and for the compression trial (ICC: 0.61 to 0.98) at the gastrocnemius (Table 2; Figure 2 and Figure 3). At the vastus lateralis, reliability was poor-to-good in the control (ICC: 0.38 to 0.80) and moderate-to-good in the compression trials (ICC: 0.46 to 0.78; Table 3).

### 3.6. Muscle Oxygenation

Muscle oxygenation reliability for the control trial at the gastrocnemius was moderate-to-good, with ICC ranging from 0.59 to 0.79 and CV of 2.7 to 6.2% (Table 4). During the compression trial, the reliability of muscle oxygenation had a CV ranging from 1.0 to 8.8% and ICC from 0.62 to 0.84 (Table 4). At the vastus lateralis, reliability was moderate-to-good in the control condition (Table 5; ICC: 0.50 to 0.64; CV: 0.6 to 4.0%) and was poor-to-good with compression (Table 5; ICC: 0.35 to 0.89; CV = 4.0 to 4.5%).

### 3.7. Muscle Deoxygenation

Table 6 demonstrates moderate reliability for muscle deoxygenation at the gastrocnemius for the control trial and moderate-to-good for the compression conditions. At the vastus lateralis, moderate-to-good reliability is demonstrated for both the control and compression conditions (Table 7).

## 4. Discussion

The purpose of the present study was to assess the between-day reliability of NIRS-determined muscle oxygenation, deoxygenation, and tissue saturation index with and without compression tights. The relative measurement error for tissue saturation, determined via an intra-class correlation, demonstrated poor-to-good reliability (ICC: 0.38 to 0.96) across all activities during the control trial. In addition, the coefficient of variation values for tissue saturation were ≤4.0%. Similarly, when wearing compression tights, tissue saturation demonstrated moderate-to-good reliability (ICC: 0.46 to 0.98), with a coefficient of variation ranging from 2.6 to 4.5%. Muscle oxygenation and deoxygenation demonstrated moderate-to-good reliability (ICC: 0.40 to 0.81), with coefficient of variation values ranging from 0.6 to 6.2% across all activities in the control trial. Compression tights trials generated moderate-to-good reliability (ICC: 0.50 to 0.93), with coefficient of variation values ranging from 0.6 to 9.9%. There were no differences in heart rate, blood pressure, pulse oximetry, or RPE during the control or compression tights trials (*p* > 0.05).

Previous research investigating the reliability of NIRS during and after exercise has shown that reliability varies with exercise mode and intensity, muscle group, NIRS device, and training status of participants [16,19,20,32]. Therefore, it is imperative that reliability is determined for the specific NIRS-derived variables during protocols that will be used to assess the effect of any intervention. In the control condition, for tissue saturation index at the gastrocnemius, the systematic bias (LOA) ranged from 0.1 to 1.7% (LOA: 4.4 to 10.3%), with coefficient of variation of 1.7 to 4.0%. At the vastus lateralis, in the control condition, systematic bias (LOA) for tissue saturation was 0.4 to 2.4% (LOA: 5.2–6.9%) and coefficient of variation values were 1.9 to 2.6%. This compares favourably with previous research that has reported reliability of NIRS for the measurement of the tissue saturation index. Coefficient of variation during handgrip exercise was 8.3 to 24.3% [16], during bicep isometric contractions was 6.9 to 37.7% [17], during knee extension was 1.6 to 2.4% [20], and during back squats was ~6.1 to 10% [23]. There has been limited research examining the reliability of NIRS to measure tissue saturation index during multi-joint movement such as running and cycling. During cycling, reliability of TSI has been shown to range between 7.8 and 12.4% [22]. Thus, the reliability of NIRS for the measurement of TSI during supine and seated rest, walking, and jogging appears very good. Similarly, for muscle oxygenation and deoxygenation measurements during the control trial, NIRS demonstrated moderate-to-good reliability. For muscle oxygenation, the coefficient of variation was 0.6 to 6.2% and for deoxygenation was 1.1 to 5.4%, demonstrating greater reliability than previous research [23]. In summary, NIRS demonstrates good reliability for the assessment of tissue saturation and muscle oxygenation and deoxygenation whilst wearing normal sporting attire. The superior reliability reported in the present study compared to previous research is likely due to the recent advancements in NIRS technology and data processing, including filtering, specific for its use in sports science research.

Despite the fact that adding pressure to an NIRS sensor has been suggested to affect the reliability of NIRS measurements, [24] by altering the depth of the measurement, to our knowledge no research has sought to establish reliability when wearing compression clothing. This is surprising since several studies have used NIRS to assess the effects of compression on muscle oxygenation and haemodynamics [6,33,34]. The present study showed that despite pressure being added to the NIRS device via compression clothing, it is reliable for the assessment of tissue saturation and muscle oxygenation and deoxygenation. At the gastrocnemius, coefficient of variation for tissue saturation index was 3.0 to 4.5%, muscle oxygenation was 1.0 to 8.8%, and deoxygenation was 4.0 to 9.9%. Similar values were recorded at the vastus lateralis, with coefficient of variation of 2.6 to 3.5% for tissue saturation index, 4.0 to 4.5% for oxygenation, and 0.6 to 3.1% for deoxygenation. One study has assessed the effect of pressure on NIRS reliability at the vastus lateralis at rest [35]. Using a thigh blood pressure cuff, pressure was applied to the NIRS sensor in 5 mmHg increments from 5 to 30 mmHg. A decrease in tissue saturation index was reported with increasing pressure, with very different slopes reported for the two devices tested (MOXY and PortaMon). The PortaMon demonstrated a shallow slope with <3% difference in tissue saturation across the different pressures, whereas it differed by >10% with the MOXY, suggesting that the MOXY should only be used if the pressure applied can be controlled.

In the present study, while systematic bias, limits of agreement, and coefficient of variation values demonstrated good reliability, it should be recognised that the variation was greater when wearing tights than without. This emphasises the importance of assessing reliability of NIRS before employing the technique in research studies or determining reliability as an integral part of the study.

NIRS has been employed in numerous exercise- and sport-related research studies in recent years. It is therefore useful to determine if the reliability or measurement error reported for NIRS in the present study is lower than changes measured during sport science research. For example, muscle tissue saturation has been observed to decrease by 1.3% for every 1 m·min^−1^ increase in climbing speed [36] and decrease from 79.7 to 62.0% following intermittent cycling to exhaustion [37]. In addition, during passive and active recovery between high-intensity intermittent cycling efforts, changes in tissue saturation index ranged from 3 to 25% and muscle oxygenation ranged from 10 to 75% [38]. At rest, young adults had lower tissue saturation index than middle aged men (73.2 vs. 68.6%) and it decreased by 17.4 and 22.2% during resistance exercise [39]. Systematic bias of <3.5% and CV <4.5% for TSI at the gastrocnemius and vastus lateralis suggest that NIRS is reliable enough to detect a range of changes seen in sports science research both at rest and during exercise.

In summary, it is imperative to determine the reliability or measurement error of NIRS before utilising it during sports science research. Despite the widespread use of NIRS within sports science, there are few published reliability studies and none that have assessed reliability whilst wearing compression garments. The present study demonstrated good reliability for all activities with small systematic bias, limits of agreement, and coefficient of variation values. When wearing bespoke compression tights, reliability was slightly poorer, therefore emphasising the importance of determining the measurement error of NIRS specifically for each study. To conclude, NIRS is sufficiently reliable to detect changes in TSI, muscle oxygenation, and deoxygenation during a range of passive and exercise activities when wearing normal sporting attire or compression-wear.

## Figures and Tables

**Figure 1 sports-11-00023-f001:**
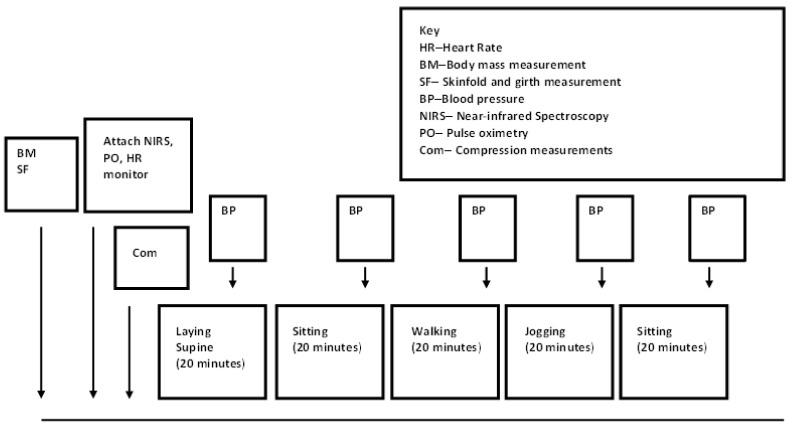
Schematic view of experimental protocol used during each trial.

**Figure 2 sports-11-00023-f002:**
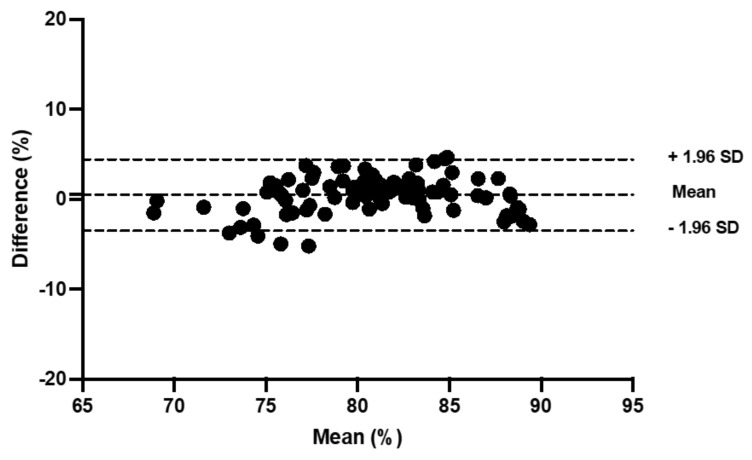
Bland and Altman plot of the sitting period post-jogging in the control trial at the gastrocnemius.

**Figure 3 sports-11-00023-f003:**
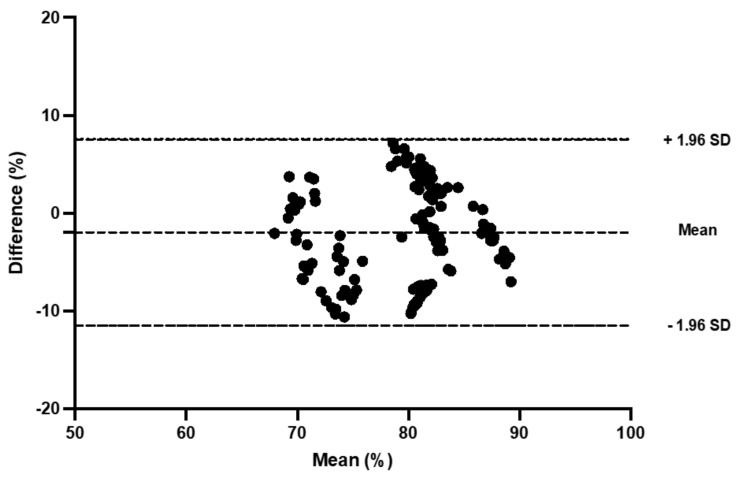
Bland and Altman plot of the sitting period post-jogging in the compression tights trial at the gastrocnemius.

**Table 1 sports-11-00023-t001:** Skinfold and girth measurements for the control and compression trials (mean ± SD).

	Gastrocnemius		Vastus Lateralis	
	Skinfold (mm)	Girth (cm)	Skinfold (mm)	Girth (cm)
Control Trial 1	9.0 ± 2.4,	39.5 ± 4.3	11.3 ± 5.1	57.0 ± 6.1
Control Trial 2	11.3 ± 5.1	40.1 ± 6.1	11.6 ± 3.1	56.5 ± 5.2
Compression Tights Trial 1	9.2 ± 4.0	40.5± 5.9	12.6 ± 5.2	54.7± 5.7
Compression Tights Trial 2	9.5 ± 4.5	38.9 ± 4.7	11.3 ± 3.8	56.7 ± 5.3

**Table 2 sports-11-00023-t002:** Tissue saturation index (%) during the control (C) and compression tights (CT) trials at the gastrocnemius (mean ± SD).

	Control 1 (%)	Control 2 (%)	Systematic Bias (C) (%)	Bland and Altman 95% LoA	ICC	CV (%)	Compression Tights 1 (%)	Compression Tights 2 (%)	Systematic Bias (CT) (%)	Bland and Altman 95% LoA	ICC	CV (%)
Supine	76.2. ± 3.5	76.6 ± 7.0	0.3	−9.9, 10.6	0.56	3.8	73.9 ± 6.7	73.1 ± 4.3	−0.7	−9.0, 7.4	0.73	3.0
Sitting	73.4 ± 6.2	75.2 ± 6.9	1.7	−3.2 6.7	0.92	1.9	73.2 ± 8.7	71.4 ± 6.7	−1.7	−13.4, 9.8	0.98	4.5
Walking	72.0 ± 7.2	71.6 ± 8.0	−0.4	−8.1, 7.2	0.87	4.0	70.9 ± 8.0	69.7 ± 5.6	−1.1	−13.2, 10.9	0.61	4.5
Jogging	67.7 ± 7.6	67.9 ± 7.6	0.2	−5.8, 6.2	0.96	2.2	63.9 ± 6.3	67.5 ± 7.0	3.5	−5.6, 12.8	0.80	4.5
Sitting	80.8 ± 4.6	81.3 ± 4.9	0.5	−3.5, 4.4	0.91	1.4	80.5 ± 5.9	78.5 ± 6.0	−1.9	−11.5, 7.5	0.67	3.5

ICC = Intra-class correlation, CV = coefficient of variation, C = control trial, CT = compression tights trial.

**Table 3 sports-11-00023-t003:** Tissue saturation index (%) during the control (C) and compression tights (CT) trials at the vastus lateralis (mean ± SD).

	Control 1 (%)	Control 2 (%)	Systematic Bias (C) (%)	Bland and Altman 95% LoA	ICC	CV (%)	Compression Tights 1 (%)	Compression Tights 2 (%)	Systematic Bias (CT) (%)	Bland and Altman 95% LoA	ICC	CV
Supine	73.3 ± 1.6	74.4 ± 3.0	1.0	−4.3, 6.5	0.38	2.0	76.2 ± 3.5	76.2 ± 3.5	0.0	−7.0, 7.0	0.47	2.6
Sitting	70.7 ± 4.1	71.5 ± 4.1	0.8	−4.3, 6.0	0.80	1.9	73.3 ± 5.0	73.9 ± 4.5	0.6	−8.7, 9.9	0.51	3.4
Walking	72.9 ± 4.1	73.7 ± 4.9	0.8	−5.3, 6.9	0.77	2.2	74.4 ± 5.2	74.7 ± 5.5	0.2	−7.2, 7.7	0.75	2.7
Jogging	71.6 ± 5.1	69.1 ± 5.2	−2.4	−8.8, 3.9	0.80	2.4	65.4 ± 5.9	65.0 ± 6.8	−0.8	−9.2, 7.6	0.78	3.1
Sitting	72.4 ± 5.3	70.9 ± 5.8	−0.4	−7.2, 6.5	0.79	2.6	75.3 ± 3.8	75.6 ± 5.3	0.3	−9.2, 9.9	0.46	3.5

ICC = Intra-class correlation, CV = coefficient of variation, C = control trial, CT = compression tights trial.

**Table 4 sports-11-00023-t004:** Oxyhaemoglobin responses from the baseline during the control (C) and compression tights (CT) trials at the gastrocnemius (mean ± SD).

	Control 1 (ΔµM)	Control 2 (ΔµM)	Systematic Bias (C) (ΔµM)	Bland and Altman 95% LoA (ΔµM)	ICC	CV (%)	Compression Tights 1 (ΔµM)	Compression Tights 2 (ΔµM)	Systematic Bias (CT) (ΔµM)	Bland and Altman 95% LoA (ΔµM)	ICC	CV (%)
Supine	−2.1 ± 6.5	0.5 ± 5.5	2.6	−6.2, 9.6	0.68	3.5	0.9 ± 9.4	−0.7 ± 5.7	−1.6	−12.8, 10.0	0.62	1.0
Sitting	13.1 ± 8.4	13.2 ± 6.6	0.1	−7.8, 7.9	0.79	3.6	1.7 ± 12.1	4.7 ± .8.0	2.4	−8.7, 12.6	0.83	1.2
Walking	1.3 ± 5.8	0.5 ± 5.0	−0.9	−5.6, 5.1	0.67	3.3	0.5 ± 9.7	−0.6 ± 19.1	−0.1	−17.9, 17.8	0.73	8.2
Jogging	−0.9 ± 8.4	−0.6 ± 9.7	0.2	−12.7, 13.0	0.68	6.2	−5.6 ± 9.9	−3.6 ± 19.3	2.0	−19.2, 22.4	0.70	8.8
Sitting	21.9 ± 9.6	26.5 ± 5.1	4.5	−8.1, 12.2	0.59	2.7	18.3 ± 16.9	7.4 ± 18.9	−11.0	−26.4, 11.9	0.84	7.5

ICC = Intra-class correlation, CV = coefficient of variation, C = control trial, CT = compression tights trial.

**Table 5 sports-11-00023-t005:** Oxyhaemoglobin responses from the baseline during the control (C) and compression tights (CT) trials at the vastus lateralis (mean ± SD).

	Control 1 (ΔµM)	Control 2 (ΔµM)	Systematic Bias (C) (ΔµM)	Bland and Altman 95% LoA (ΔµM)	ICC	CV (%)	Compression Tights 1 (ΔµM)	Compression Tights 2 (ΔµM)	Systematic Bias (CT) (ΔµM)	Bland and Altman 95% LoA (ΔµM)	ICC	CV (%)
Supine	−2.4± 4.3	−3.5± 4.5	−1.1	−8.1, 6.5	0.5	0.6	−3.0 ± 8.2	−1.0 ± 3.3	2.1	−8.2, 10.6	0.50	4.5
Sitting	−0.8 ± −3.5	−1.0 ± 7.6	−0.1	−8.6, 8.3	0.5	3.1	3.3 ± 10.3	5.2 ± 12.7	2.0	−7.8, 10.4	0.89	4.5
Walking	5.2 ± 5.9	−5.7 ±3.5	−10.8	−15.5, 7.4	0.64	3.0	1.2 ± 3.8	−0.1 ± 5.9	−1.4	−8.1, −6.9	0.35	4.1
Jogging	1.9± 6.4	−2.6 ±5.1	−4.61	−11.6, 7.0	0.52	3.8	1.3 ± 4.8	−1.3 ± 7.9	−2.6	−11.7, −8.3	0.55	4.4
Sitting	5.7 ± 5.5	−0.1± 8.4	−5.50	−17.8, 6.8	0.61	4.0	1.8 ± 7.0	3.4 ± 5.5	1.7	−7.8, 9.8	0.61	4.0

ICC = Intra-class correlation, CV = coefficient of variation, C = control trial, CT = compression tights trial.

**Table 6 sports-11-00023-t006:** Deoxyhaemoglobin responses from the baseline during the control (C) and compression tights (CT) trials in the gastrocnemius (mean ± SD).

	Control 1 (ΔµM)	Control 2 (ΔµM)	Systematic Bias (C) (ΔµM)	Bland and Altman 95% LoA (ΔµM)	ICC	CV (%)	Compression Tights 1 (ΔµM)	Compression Tights 2 (ΔµM)	Systematic Bias (CT) (ΔµM)	Bland and Altman 95% LoA (ΔµM)	ICC	CV (%)
Supine	−9.3 ± 4.4	−8.1 ± 2.7	1.1	−5.7, 7.1	0.43	2.1	−11.5 ± 8.6	−11.0 ± 9.4	0.3	−12.5, 14.3	0.82	4.0
Sitting	2.3 ± 4.6	1.8 ± 4.8	−0.5	−7.5, 6.8	0.57	2.9	−2.2 ± 9.7	−2.1 ± 9.4	0.1	−12.2, 12.5	0.71	5.3
Walking	−5.1 ± 2.6	−5.6 ± 3.5	−0.3	−4.6, 4.1	0.57	1.1	3.6 ± 9.0	−2.0 ± 7.8	−1.9	−15.0, 8.7	0.65	5.1
Jogging	0.1 ± 7.4	1.7 ± 6.1	1.6	−9.7, 11.7	0.52	1.6	10.3 ± 12.2	5.5 ± 12.0	−4.8	−16.0, 10.6	0.76	6.2
Sitting	−2.7 ± 4.0	−2.9 ± 3.4	−0.3	−5.0, 4.8	0.61	5.4	0.0 ± 16.7	−2.0 ± 9.9	−2.1	−25.0, −21.9	0.53	9.9

ICC = Intra-class correlation, CV = coefficient of variation, C = control trial, CT = compression tights trial.

**Table 7 sports-11-00023-t007:** Deoxyhaemoglobin responses from the baseline during the control (C) and compression tights (CT) trials in the vastus lateralis (mean ± SD).

	Control 1 (ΔµM)	Control 2 (ΔµM)	Systematic Bias (C) (ΔµM)	Bland and Altman 95% LoA (ΔµM)	ICC	CV (%)	Compression Tights 1 (ΔµM)	Compression Tights 2 (ΔµM)	Systematic Bias (CT) (ΔµM)	Bland and Altman 95% LoA (ΔµM)	ICC	CV (%)
Supine	−3.3 ± 2.0	−4.4 ± 2.9	−1.1	−5.2, 3.5	0.50	1.8	−1.3 ± 7.3	−0.8 ± 7.2	0.6	−4.4, 5.0	0.93	0.6
Sitting	−1.3 ± 1.7	−1.3 ± 4.6	0.0	−5.2, 5.3	0.42	2.7	−1.8 ± 3.7	−0.7 ± 2.9	0.9	−3.0, 4.7	0.79	1.5
Walking	−2.1 ± 2.0	−2.6 ± 5.9	−0.5	−7.4, 6.9	0.40	3.5	−3.3 ± 3.5	−1.8 ± 2.9	1.5	−4.2, 5.1	0.82	1.6
Jogging	2.2 ± 5.5	−1.9 ± 7.0	−4.1	−10.6, 4.3	0.81	2.8	−1.6 ± 5.4	0.0 ± 4.5	1.4	−2.7, 5.0	0.89	1.6
Sitting	−3.5 ± 2.8	−3.7 ± 4.6	−0.1	−6.0, 5.8	0.56	2.5	−3.1 ± 5.4	−2.8 ± 3.8	0.2	−7.5, 7.9	0.53	3.1

ICC = Intra-class correlation, CV = coefficient of variation, C = control trial, CT = compression tights trial.

## Data Availability

Data are available via appropriate request to the authors and institutional ethical committee, until its destruction in line with the data management plan.

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
