# Peer review of "Reliability of Near-Infrared Spectroscopy with and without Compression Tights during Exercise and Recovery Activities"

_sports, 2023, doi:10.3390/sports11020023_

Round 1

Reviewer 1 Report

Dear Authors,

The study aims to assess the between-day reliability of NIRS for the determination of muscle oxygenation, deoxygenation and tissue saturation index at the gastrocnemius and vastus lateralis with and without compression tights.

In general, paper sufficiently advances knowledge to merit publication, the approach is relevant.

The authors report on an interesting topic that is relevant from both a theoretical and practical perspective to the readers of Sports.

Authors have in introduction presented with references supported problem of their study.

Moreover, the introduction is written clear and concise with the study aim in mind.

Methodology has been adequate to meet the objectives.

Content of the paper is within the scope of the journal and presentation is sufficiently clear and precise. Finally, Conclusions are justified by the experimental evidence.

Congratulations to you and your co-authors in meeting the very high standard of quality that is required for publication in this Journal.

Kind regards

Author Response

Many thanks to the reviewer for their positive comments and taking the time to the review the paper. We very much appreciate your efforts.

Reviewer 2 Report

First of all, I would like to congratulate the researchers for their efforts.

The research design is correct, but I think the logic is wrong.

-If an equipment provides reliable data and has proven it in a few sessions, the pressure is unlikely to have an effect on it.

-For example, if the scale has a weighing capacity of 200kg and my 100kg measurement shows 100kg each time, this is reliable. It will be necessary to investigate their validity rather than reliability so that I can control them at different weights.

-In conclusion, I think the research design is unreasonable.

-References do not comply with journal guidelines.

-Analyzes need to be further developed and presented to readers with more understandable visuals.

Author Response

First of all, I would like to congratulate the researchers for their efforts.

Thank you for taking the time to review the paper. The additions we have made have improved the clarity of the paper.

 The research design is correct, but I think the logic is wrong.

-If an equipment provides reliable data and has proven it in a few sessions, the pressure is unlikely to have an effect on it.

Please see the response below relating to why pressure on the device may impact its reliability and thus is the focus of this study.

-For example, if the scale has a weighing capacity of 200kg and my 100kg measurement shows 100kg each time, this is reliable. It will be necessary to investigate their validity rather than reliability so that I can control them at different weights.

-In conclusion, I think the research design is unreasonable.

Many thanks for these comments. Previous research has suggested that pressure on the NIRS optodes could impact upon the validity and thus reliability of the measurements (Hamaoka et al. 2011). Despite this, no studies have examined how the pressure may impact reliability and this is the key novel aspect of this study. Clearly this is important for any study that was to investigate compression garments but it may also be very relevant for other studies that use NIRS as many studies apply a dark bandage on top of the optode to keep light out but do not measure the pressure exerted on the device. Therefore, knowing whether pressure does affect the reliability of NIRS is essential for all researchers using NIRS.

To improve the clarity of this in the introduction, we have added an additional sentence following on from our original sentence where we outlined that pressure has been suggested to impact upon the reliability of the NIRS measurements.

Previous section in the introduction:

In addition, applying pressure to the muscle and NIRS optode, which would occur when wearing compression garments, can alter the overall reliability of NIRS and must be evaluated [23].

New section in the introduction:

In addition, applying pressure to the muscle and NIRS optode, which would occur when wearing compression garments, can alter the overall reliability of NIRS and must be evaluated [24]. Variations in pressure applied to the NIRS device can alter the pathlength between the muscle and the optode subsequently impacting the signal quality [24].

-References do not comply with journal guidelines.

 Many thanks for highlighting this. We have now downloaded the updated MDPI Endnote style and used this throughout the paper.

-Analyzes need to be further developed and presented to readers with more understandable visuals.

We have added in 2 Bland and Altman plots to provide a visual example to the reader of the data in the Tables.

Reviewer 3 Report

Review of the scientific article entitled “Reliability of Near Infrared Spectroscopy With and Without Compression Tights During Exercise and Recovery Activities”

The aim of this paper was to assess the reliability of NIRS with and without compression tights. The use of compression has become popular in sport, so it provides very valuable and novel information in the field of sports science for performance and healthy fitness. This manuscript is very well written and oriented. The manuscript contributes substantially to improve the analysis of the use of NIRS to study changes in muscle oxygenation. We believe that some minor changes are needed. Besides, I still have some doubts to clarify.

Line 12: Please delete “; mean ± SD”, and add the average and SD value of “body mass index”.

Line 13: Please delete the comma after “tights”

Line 14: It is not clear that 20 minutes of jogging are performed. Will the measurements be analyzed during jogging, or is it only during recovery; the sitting is the recovery from jogging? Please clarify

Lines 18 - 19:  Please change the separation "-" in all systematic bias ranges and coefficients of variation (CV) to "to", such as "from XX to YY". Please make these changes throughout the manuscript.

Lines 43 - 44: Compression garments reduced blood flow during recovery?,

Line 53: There is a misquote that has not been referenced: “(Jones et al., 2016)” 

Line 53-57: Please improve the wording of the sentence.

Line 60: Apparently there is an error in the quotation: “Balas, Kodejska [15]”

Line 63: Please change “-” for “to”. The same for 66 and 68.

Line 85: Please delete “mean SD” and add the average and SD of “body mass index”.

Line 86: Please add the ethics committee approval code.

Line 100: How long was the jogging recovery, was it after 20 min of jogging, was it after 20 min of jogging? Can you explain how much recording time you did for each analysis, was it an average recording time of 20 min for all? If you measured jogging recovery, could you make it clearer in the results tables.

Line 136-137: TSI is an absolute measure that represents tissue oxygenation? Could you add how it is obtained (e.g., equation)? In the same section you could put the units of the biological chromophores and TSI. With this, it would not be necessary to have the sentence in lines 156 - 157: "Tissue saturation 156 index is an absolute value and is expressed as percentage saturation", please delete. Finally, add in the studied changes also TSI (lines 155 - 156).

Line 159: Could you place the sentence: "Statistical analysis was performed using SPSS version 24.0 (SPSS Inc., Chicago, IL, 159 USA)" at the end of this section (line 166)?

Line 166: Could you place the sentence: "Data are presented as mean ± SD, at the beginning of this section (line 159)?

Line 143: Please go to section “Physiological and Perceptual Measurements” just before the paragraph “Near-infrared Spectroscopy measurements” (lines 122-123).

Line 123 - 142: It is not clear if the changes will be measured during the 20 minutes, or at the beginning and at the end, please explain.

Line 187: It says that the CV was less than 7.2 in both groups, however on line 188 it says that during Jogging it was 7.5 for CT.

Author Response

Reviewer 3

Review of the scientific article entitled “Reliability of Near Infrared Spectroscopy With and Without Compression Tights During Exercise and Recovery Activities”

The aim of this paper was to assess the reliability of NIRS with and without compression tights. The use of compression has become popular in sport, so it provides very valuable and novel information in the field of sports science for performance and healthy fitness. This manuscript is very well written and oriented. The manuscript contributes substantially to improve the analysis of the use of NIRS to study changes in muscle oxygenation. We believe that some minor changes are needed. Besides, I still have some doubts to clarify.

Many thanks for taking the time to review the paper and making suggestions for improvement. We have addressed these below and made the requested changes which has improved the manuscript.

Line 12: Please delete “; mean ± SD”, and add the average and SD value of “body mass index”.

The BMI data has been added.

Line 13: Please delete the comma after “tights”

Comma has been removed.

Line 14: It is not clear that 20 minutes of jogging are performed. Will the measurements be analyzed during jogging, or is it only during recovery; the sitting is the recovery from jogging? Please clarify

Yes, this was confusing. We have tried to make it clearer by rewording the sentence as below:

During each trial, participants completed 20 minutes each of laying supine, sitting, walking (4km.h-1), jogging and sitting following the jogging.

Lines 18 - 19:  Please change the separation "-" in all systematic bias ranges and coefficients of variation (CV) to "to", such as "from XX to YY". Please make these changes throughout the manuscript.

This change has been made throughout the manuscript.

Lines 43 - 44: Compression garments reduced blood flow during recovery?,

Interestingly the Sperlich et al (2013) study demonstrates a reduction in blood flow following exercise with compression. There conclusion being “These results demonstrate that wearing compression shorts with 37 mmHg of external pressure reduces blood flow both in the deep and superficial regions of muscle tissue during recovery from high intensity exercise”.

The authors suggested this difference from other studies may relate to pressure levels and design. It provides another example of the equivocal nature of the research in compression garments.

Line 53: There is a misquote that has not been referenced: “(Jones et al., 2016)” 

Thanks for spotting this. It has now been referenced correctly.

Line 53-57: Please improve the wording of the sentence.

This has been altered to try to improve the flow. The new sentence is:

NIRS can therefore be applied to studies examining the physiological responses to compression clothing to determine blood flow and oxygenation. However, to be used during exercise compression studies, the reliability of NIRS must be determined as reliability appears to be specific to the exercise type and intensity.

Line 60: Apparently there is an error in the quotation: “Balas, Kodejska [15]”

These has been updated appropriately in Endnote.

Line 63: Please change “-” for “to”. The same for 66 and 68.

This change has been made here and throughout the rest of the manuscript.

Line 85: Please delete “mean SD” and add the average and SD of “body mass index”.

The BMI data has been added.

Line 86: Please add the ethics committee approval code.

The approval number has been added. 425.

Line 100: How long was the jogging recovery, was it after 20 min of jogging, was it after 20 min of jogging? Can you explain how much recording time you did for each analysis, was it an average recording time of 20 min for all? If you measured jogging recovery, could you make it clearer in the results tables.

The jogging was at a speed that the individuals would go for a ‘recovery’ jog at. Immediately following this 20 minute jog the participants sat down for a further 20 minutes to assess any changes following this jog (sitting). We have tried to reword this section slightly to make it clearer.

Line 136-137: TSI is an absolute measure that represents tissue oxygenation? Could you add how it is obtained (e.g., equation)? In the same section you could put the units of the biological chromophores and TSI. With this, it would not be necessary to have the sentence in lines 156 - 157: "Tissue saturation 156 index is an absolute value and is expressed as percentage saturation", please delete. Finally, add in the studied changes also TSI (lines 155 - 156).

We have added in the equation for TSI and the units and removed the sentence at 156-157.

Line 159: Could you place the sentence: "Statistical analysis was performed using SPSS version 24.0 (SPSS Inc., Chicago, IL, 159 USA)" at the end of this section (line 166)?

The sentence has been moved.

Line 166: Could you place the sentence: "Data are presented as mean ± SD, at the beginning of this section (line 159)?

Again, this sentence has been moved as suggested.

Line 143: Please go to section “Physiological and Perceptual Measurements” just before the paragraph “Near-infrared Spectroscopy measurements” (lines 122-123).

This section has been moved as suggested.

Line 123 - 142: It is not clear if the changes will be measured during the 20 minutes, or at the beginning and at the end, please explain.

Thanks for pointing this out. We have added additional information into the data processing section to clarify this. We averaged per minute producing 20 data points per 20 minute activity.

Line 187: It says that the CV was less than 7.2 in both groups, however on line 188 it says that during Jogging it was 7.5 for CT.

Well spotted. We have updated that.

Round 2

Reviewer 2 Report

Congratulations to the authors. It can also be published by other reviwers and editor if appropriate.